



# Low-altitude frequency-banded equatorial emissions observed below the electron cyclotron frequency

**Mohammed Y. Boudjada**[1], **Patrick H. M. Galopeau**[2], **Sami Sawas**[3], **Valery Denisenko**[4,5], **Konrad Schwingenschuh**[1], **Helmut Lammer**[1], **Hans U. Eichelberger**[1], **Werner Magnes**[1], **and Bruno Besser**[1]

[1]Space Research Institute, Austrian Academy of Sciences, Graz, Austria
[2]LATMOS-CNRS, Université Versailles Saint-Quentin-en-Yvelines, Guyancourt, France
[3]Institute of Communications and Wave Propagation, Graz University of Technology, Graz, Austria
[4]Institute of Computational Modelling, Russian Academy of Sciences, Krasnoyarsk, Russia
[5]Siberian Federal University, Krasnoyarsk, Russia

**Correspondence:** Mohammed Y. Boudjada (mohammed.boudjada@oeaw.ac.at)

**Abstract.** The ICE (Instrument Champ Électrique) experiment on board the DEMETER (Detection of Electro-Magnetic Emissions Transmitted from Earthquake Regions) satellite recorded frequency-banded wave emissions below the electron cyclotron frequency, with band spacing $\gtrsim$ frequency low-hybrid resonance, in the vicinity of the magnetic equatorial plane. Those radiations were observed in the beginning of the year 2010 on the night side of Earth and rarely on the day side. We distinguish two components: one appears as frequency bands continuous in time between a few kilohertz and up to 50 kHz, and the other one is from 50 to 800 kHz. The first component exhibits positive and negative frequency drift rates in the Southern Hemisphere and Northern Hemisphere, at latitudes between 40 and 20°. The second one displays multiple spaced frequency bands. Such bands mainly occur near the magnetic equatorial plane with a particular enhancement of the power level when the satellite latitude is close to the magnetic equatorial plane. We show in this study the similarities and the discrepancies between the non-free-space DEMETER frequency-banded emissions and the well-known free-space terrestrial kilometric radiation. The hollow cones of the DEMETER frequency-banded wave emissions are oriented towards Earth's ionosphere. We suggest that the source region is localized in regions poleward of the plasmapause where the ratio of the plasma frequency to gyro-frequency is bigger than one.

## 1 Introduction

A variety of radio waves were detected in the near-Earth space environment in the 1970s. The first type of waves were observed at frequencies below 100 and up to 30 kHz (Brown, 1973) and even lower between 5 and 20 kHz (Gurnett and Shaw, 1973). These two types of emission belong to a single non-thermal continuum spectrum, one "trapped" and the other "escaping" (Gurnett, 1975). High-resolution spectrograms also made evident the presence of numerous narrow-band emissions for the escaping component (Kurth et al., 1981). Later on, the Cluster tetrahedral configuration of four identical satellites allowed for the analysis of a specific type of a nonthermal continuum (Décréau et al., 2001). A direction-finding technique, based on antenna spin modulation, allowed for localizing the source regions in the plasmapause (Décréau et al., 2004), confirming previous Geotail observations (Hashimoto et al., 1999). Grimald et al. (2008) showed in the nonthermal emissions the presence of spectral peaks organized as several banded emissions with a frequency interval nearby the gyrofrequency at the sources. The considered event was recorded on 30 December 2003, in the Southern Hemisphere and Northern Hemisphere before and after plasmapause boundaries. All satellites display a similar behaviour with arranged peaking times (i.e. C1, C2, C3 and then C4) corresponding to the satellite ordering along the "line of pearls configuration". Authors suggested a stable beam of a limited cone angle. Further polarization investigations of such a type of banded emissions by Grimald and

Santolik (2010) led to the conclusion that the observed polarization excludes the presence of the Langmuir mode and the ordinary mode. Details on the wave spectral signature were also investigated by El-Lemdani Mazouz et al. (2009), particularly the splitting in fine frequency bands. Another type called "nonthermal continuum patches" was found to occur within a relatively short time and over a wide frequency range (Grimald et al., 2011). Authors showed that plasmaspheric "patch" events represent 25 % of the total nonthermal emissions recorded in 1 year.

Also space observations provided by the IMAGE (Imager for Magnetopause-to-Aurora Global Exploration) satellite (Burch, 2000) allowed for a better investigation of the inner plasmasphere. The Radio Plasma Imager (RPI) was designed to use a radio sounding technique leading the reception of echoes from remote plasma regions. Emitted pulses can propagate in the $Z$ mode and also the whistler mode (Carpenter et al., 2003). Hence signals detected at frequencies below the local upper-hybrid frequency $f_{uh} = (f_p^2 + f_g^2)^{1/2}$ could propagate in the whistler and $Z$ modes. Here $f_p$ and $f_g$ are the plasma frequency and the gyro-frequency, respectively. Sonwalkar et al. (2004) showed that the $f_p/f_g$ ratio leads to the sounding of different regions of the plasmasphere. Hence the condition $f_p/f_g > 1$ allows for sounding below 2000 and above 4000 km within Earth's plasmasphere. In regions poleward of the plasmapause, the second condition prevails, i.e. $f_p/f_g < 1$. Similar plasma conditions were derived from the theoretical approach by Goertz and Strangeway (1995) using the Appleton–Hartree dispersion relation.

In this paper, we analyse the frequency-banded radiation observed by the ICE (Instrument Champ Électrique) DEMETER (Detection of Electro-Magnetic Emissions Transmitted from Earthquake Regions) experiment in the beginning of the year 2010. The characteristics of this radiation, essentially the spectral features and the spatial occurrence, are described in Sect. 2. A discussion of the outcomes is detailed in Sect. 3, where principally our results are combined with previous ones. A summary of the main results is given in Sect. 4.

## 2 Frequency-banded wave emission

### 2.1 Overview of high-frequency ICE observations

We consider in this study the space observations provided by the DEMETER microsatellite. The aim is the analysis of particular spectral features recorded by the ICE experiment in the beginning of the year 2010, i.e. January, February and March. The ICE instrument allows for a continuous survey of the electric field over a wide frequency range, from a few hertz up to about 3.5 MHz (Berthelier et al., 2006). The electric field component is determined along the axis defined by two sensors. The satellite sun-synchronous half-orbit duration is about 40 min, covering the invariant latitude between −65 and +65°. The DEMETER satellite orbits are associated with two fixed local times, at about 10 and 22 LT. We use in this investigation the survey mode of the ICE experiment covering the frequency range between a few kilohertz and 3.5 MHz, called hereafter the HF (high-frequency) band. The radio wave emissions are alternately recorded on the day and night sides of Earth corresponding respectively to down and up half orbits. However the main radiations investigated in this paper are observed on the night side and rarely on the day side. Generally the ICE HF-band dynamic spectra allow for distinguishing three kinds of spectral emissions depending on the satellite geographical latitudes. The first one is recorded close to the sub-auroral regions at latitudes between 50 and 60°; it mainly concerns the auroral kilometric radiation described by Parrot and Berthelier (2012). The second are mainly ground-based transmitters, low-frequency (LF) radiation, appearing at mid latitudes between 50 and 20°, in both hemispheres (e.g. Parrot et al., 2009; Boudjada et al., 2017).

The third kind of emission is a frequency-banded wave radiation occurring in the vicinity of the equatorial magnetic plane at low latitudes. Hereafter we focus on the analysis of the banded radiation, in particular the spectral characteristics, the magnetic latitude and the power intensity occurrence. Also the dependence of the power level on the frequency and the magnetic latitude is considered. We use a manual technique which consists of following and saving the frequency and the temporal evolution of the radiation with a computer mouse. The saved parameters are the observation time (UT), the frequency (kHz) and the power level ($\mu$V m$^{-1}$ Hz$^{-1/2}$). The collected points are later combined with the satellite orbital parameters like the magnetic latitude and the $L$ shell.

### 2.2 Frequency and time characteristics

The DEMETER ICE experiment detected frequency-banded emissions in the frequency range between a few kilohertz and up to 800 kHz. Two examples recorded on the night side are shown in Figs. 1 and 2. Figure 1a displays the dynamic spectrum recorded by the ICE experiment on 21 February 2010 between 13:52 and 14:12 UT. The satellite was on the late-evening sector, around 22 LT, at a distance of 665 km. In this time interval the satellite geographical coordinate varied from −18° S to +04° N in latitude and 142 to 138° in longitude. The second event shown in the first panel of Fig. 2 was also recorded at about 22 LT at a similar distance from Earth. Satellite geographical coordinates varied from −25° S to 40° N in latitude and 117 to 102° in longitude in the time interval between 14:10 and 14:26 UT. Figures 1b and 2b display a zoomed-in part of the dynamic spectrum shown in the first panels where the emission appears in the frequency range between a few kilohertz and up to 800 kHz. Note for Figs. 1b and 2b the changes in the spectral emissions before and after 50 kHz. Hence the first radiation appears as a narrow continuum with an instantaneous bandwidth of about 2 kHz at frequencies less than 50 kHz. It displays negative

*Ann. Geophys., 38, 1–10, 2020* https://doi.org/10.5194/angeo-38-1-2020

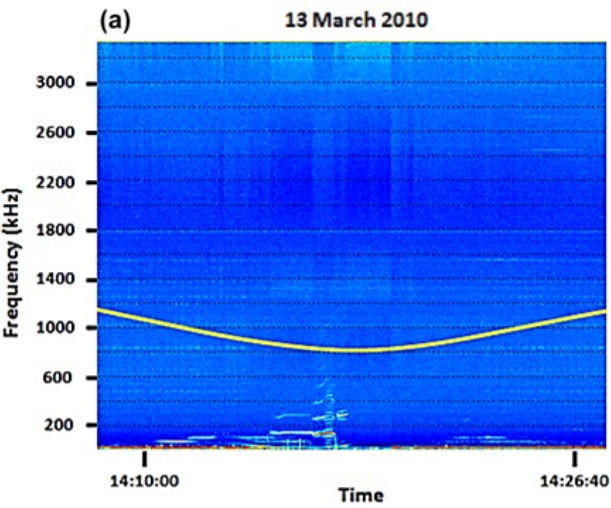

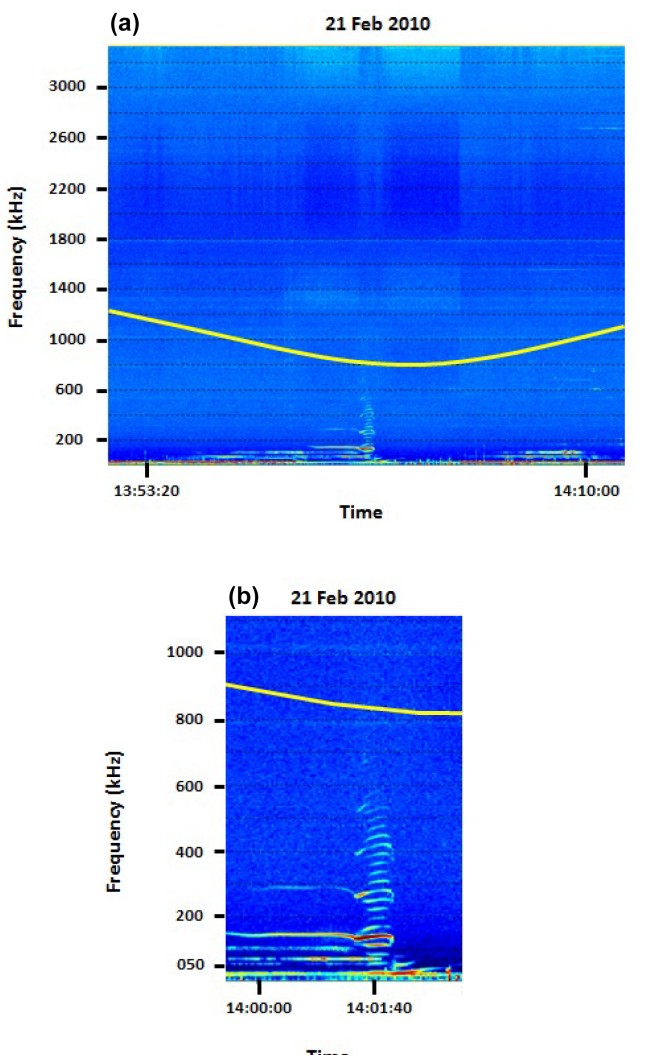

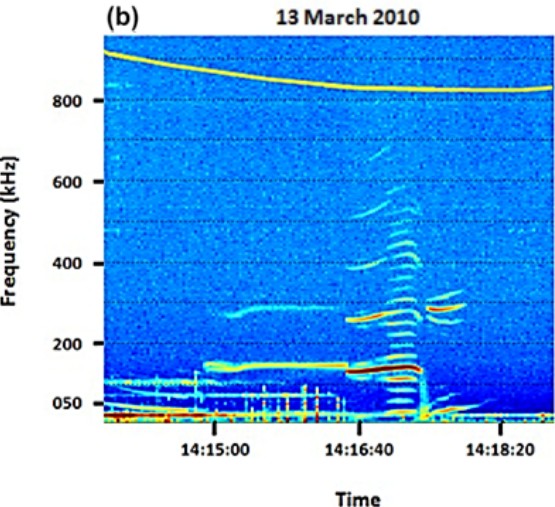

**Figure 1.** Example of frequency-banded wave emission recorded by the ICE experiment on board DEMETER on 21 February 2010. Panel (**a**) displays an overview of the dynamic spectrum in the frequency range from a few kilohertz to 3.5 MHz. Panel (**b**) shows a zoomed-in part for the event in the frequency bandwidth between a few kilohertz and 1100 kHz. The gyro-frequency is indicated by the yellow curve.

**Figure 2.** Like in Fig. 1 for an event recorded by DEMETER on 13 March 2010.

and positive frequency drifts when the satellite is approaching or leaving the equatorial plane, respectively. Its frequency drift rate is weak and in the order of 0.2 kHz s$^{-1}$. The second emission is composed of parallel narrow bands for frequencies above 50 and up to 800 kHz. The band time duration is, on average, about 1 min and decreases to less than 1 min when the emission frequency increases.

The number of parallel narrow bands is found to be different from one event to another. Hence we find, respectively, 18 and 20 parallel bands on 21 February 2010 (Fig. 1) and 13 March 2010 (Fig. 2). The spacing of the frequency band, on average, is about 30 kHz when we consider both events.

One can note that some narrow bands showed a high power level (red in Figs. 1 and 2), when they are compared to other narrow bands, and they also exhibit an extensive time duration of about a few minutes. Such enhanced narrow bands appear at 140, 270 and 540 kHz in Fig. 1 and at 130, 250, 410 and 550 kHz in Fig. 2. The enhanced banded frequencies above 200 kHz, may be considered harmonic components of a "fundamental" frequency which appears around 140 kHz and which exhibits a longer time duration. It follows that five short weak narrow-band emissions separate the basic frequency, i.e. 140 kHz, from its first harmonic around 280 kHz. Hereafter, we consider a statistical analysis of all events observed in January, February and March 2010.

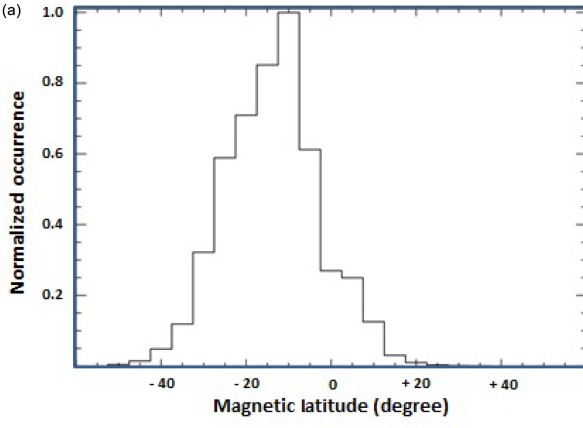

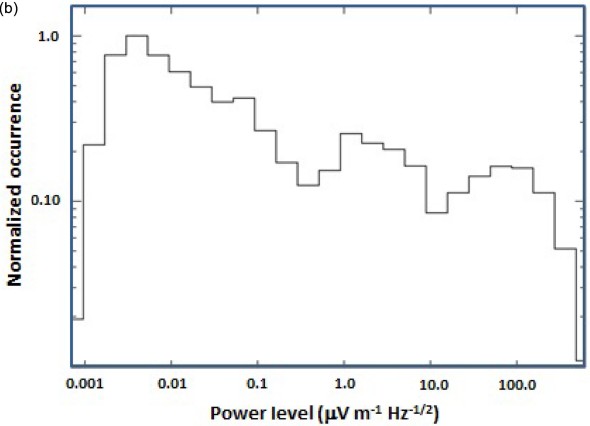

**Figure 3.** Occurrence of frequency-banded radiations in **(a)** magnetic latitude (degrees) and **(b)** power level ($\mu\text{V}\,\text{m}^{-1}\,\text{Hz}^{-1/2}$).

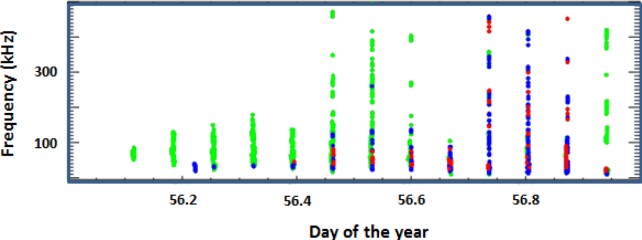

**Figure 4.** Vertical lines indicate the occurrence of the frequency-banded emissions observed on 25 February 2010. Those events were recorded on the night side of Earth with a time interval of about 1 h 35 min. Green, blue and red specify, respectively, three power level intervals, i.e. $10^{-3}$–0.7, 0.7–10 and 10–$10^{+4}\,\mu\text{V}\,\text{m}^{-1}\,\text{Hz}^{-1/2}$.

### 2.3 Magnetic latitude and power level occurrence

The frequency-banded radiation occurrences in magnetic latitude and power level are shown, respectively, in Fig. 3a and b. The main emissions were recorded when DEMETER was in the southern part of the magnetic equatorial plane. Hence the emissions are detected in the magnetic latitude range between −40 and 20°, as shown in Fig. 3a. We note a clear progressive increase of the frequency-banded emission occurrence which reaches a maximum at a magnetic latitude of −10°. More than 90 % of the radiation occurred at a range in magnetic latitude of between −30 and 0°. A sudden decrease of the occurrence is recorded when the satellite crosses the magnetic equatorial plane. Emission is found to be more extended in the Southern Hemisphere with a clear di-symmetry occurrence before and after the equatorial magnetic plane.

The power level, as displayed in Fig. 3b, covers a large interval between $10^{-3}$ and $10^{+4}\,\mu\text{V}\,\text{m}^{-1}\,\text{Hz}^{-1/2}$. More than 70 % of emissions have a level less than $1\,\mu\text{V}\,\text{m}^{-1}\,\text{Hz}^{-1/2}$ and belong mainly to the Southern Hemisphere. Above this weak power level, the occurrence of the frequency-banded emission is associated with both hemispheres. The intense power level is associated with the emission occurring mainly at lower frequency, i.e. from a few kilohertz and up to 100 kHz. We distinguish three occurrence maxima at about $5 \times 10^{-3}$, 1 and $80\,\mu\text{V}\,\text{m}^{-1}\,\text{Hz}^{-1/2}$. We separate the power level by taking into consideration the interval associated with previous maxima. Hereafter green, blue and red indicate, respectively, three power level intervals, i.e. 0.001–0.7, 0.7–10 and 10–$10^4\,\mu\text{V}\,\text{m}^{-1}\,\text{Hz}^{-1/2}$.

Frequency-banded wave emissions are regularly observed on the night side (22 LT) before and after the magnetic equatorial plane in the vicinity of Earth at a distance less than 750 km. Figure 4 displays the daily occurrence of frequency-banded emissions on 25 February 2010. We observe a periodic occurrence of the emission with a time interval of about 1 h 35 min which corresponds to a full orbit of the DEMETER microsatellite. Each vertical line is considered an "event" and corresponds to the recorded emission for a given half orbit. The occurrence per day is about 13 events in the optimal case. However from one event to another we find a variation in the frequency bandwidth and also in the power level.

### 2.4 Power level versus frequency and magnetic latitude

Figure 5 displays the power level variation versus the magnetic latitude where the colours indicate different power levels as defined in the previous subsection. The weakest intensities (less than $0.7\,\mu\text{V}\,\text{m}^{-1}\,\text{Hz}^{-1/2}$) are recorded at magnetic latitudes between −50 and +30° but much more in the Southern Hemisphere, as displayed in Fig. 5a. Structured emissions appear when the magnetic latitude is positive principally after the crossing of the magnetic equatorial plane. One can distinguish five components appearing in four frequency ranges: a few kilohertz to 50, 70 to 130, 170 to 250, 280 to 340 and 380 to 420 kHz. Those radiations are extended in magnetic latitudes in particular at low frequencies around 50 kHz and decrease at higher frequen-

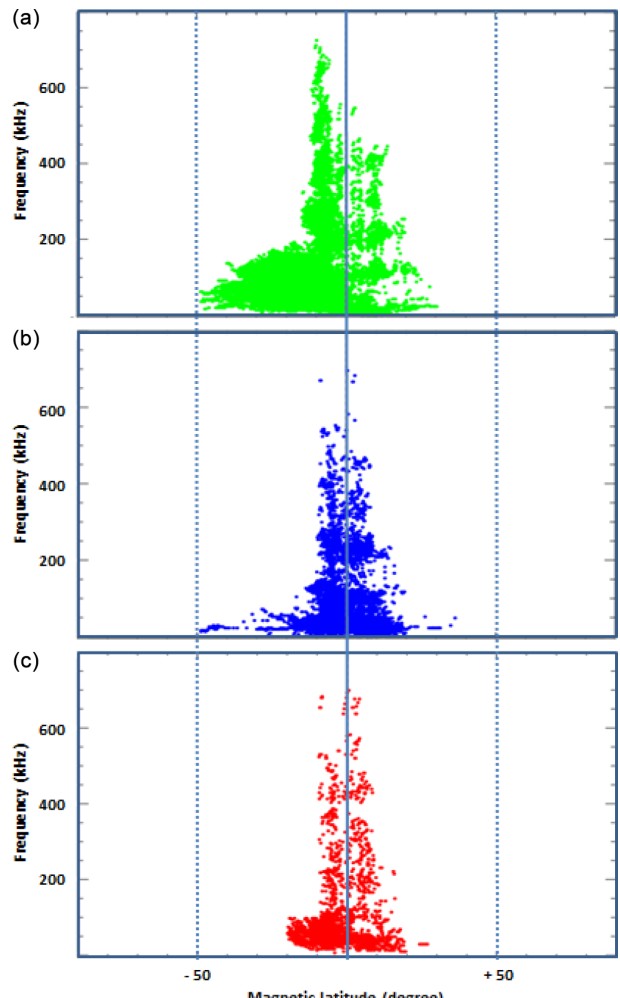

**Figure 5.** Variation of the power levels versus the frequency (vertical axis) and the magnetic latitude (horizontal axis) for all events. Colours are similar to those used in Fig. 4. Green, blue and red specify, respectively, three power level intervals, i.e. $10^{-3}$–0.7, 0.7–10 and $10$–$10^{+4}$ µV m$^{-1}$ Hz$^{-1/2}$.

cies at about 400 kHz. Frequency-banded emission is quasi-absent between those four frequency bands.

Also structured emissions are observed in the southern part of the magnetic equatorial plane at frequencies above 200 kHz in magnetic latitude between −10 and 0°, as shown in Fig. 5a. Those structures are mainly extended in frequency, contrary to those observed in the Northern Hemisphere, which extended in magnetic latitude. We distinguish four components occurring in the following frequency bands: 200–320, 320–450, 450–570 and 570–670 kHz. At frequencies lower than 200 kHz, we note a quasi-absent structured emission in the Southern Hemisphere. Radiations continuously occur in magnetic latitude between −50 and 0°. In this interval, we find a positive or negative frequency drift rate of about +3.75 or −1.25 kHz per degree when the frequency is higher or lower than 50 kHz. The emissions are mainly con-

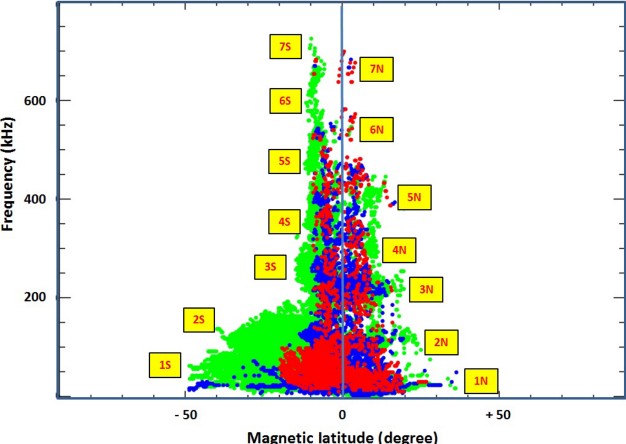

**Figure 6.** Overlapping of the three power levels displayed in Fig. 5. The spectral pattern is a "Christmas tree" with a "trunk" along the magnetic equatorial plane. We have indicated by numbers the main parts of the spectral pattern for the Southern Hemisphere and Northern Hemisphere. Table 1 lists the observational parameters associated with the investigated events.

fined to frequencies lower than 150 or 100 kHz in the southern or northern part of the magnetic equatorial plane, when the power level is between 0.7 and 10 µV m$^{-1}$ Hz$^{-1/2}$, as displayed in Fig. 5b. Above 150 kHz, the radiations only occur in the frequency bandwidth of 180 to about 250 kHz. The power level in the range $10$–$10^4$ µV m$^{-1}$ Hz$^{-1/2}$ is shown in Fig. 5c. The main emission is nearly symmetrical, distributed around the magnetic equatorial plane, between −10 and +10°, predominantly above 100 kHz. Below this limit, the radiation covers larger magnitude latitudes from −20 to about +20°.

The overlapping of the three power levels, as shown in Fig. 6, allow getting a global shape similar to a "Christmas tree" spectral pattern. We see globally that the frequency-banded emission extensively occurs at frequencies lower than 150 kHz and starts to be less confined to the magnetic equatorial plane above this frequency limit. A cut-off appears around 50 kHz, which decreases to about a few kilohertz when approaching the magnetic equatorial plane. This cut-off is characterized by a small frequency drift rate in latitude and a power level in the interval 0.7 and 10 µV m$^{-1}$ Hz$^{-1/2}$, i.e. blue boundary in Fig. 6. A second cut-off can be seen when the DEMETER satellite was in the Southern Hemisphere and absent in the Northern Hemisphere. It starts at latitudes of about −40° and disappears at −18°, when the frequency decreases from 150 to 50 kHz. We find that both cut-offs intersected at a frequency of about 50 kHz when the magnitude latitude is about −18°.

Table 1 lists the main observational parameters derived from Fig. 6. For each hemisphere, the opening angle of the beam, the frequency range and the magnetic latitude are indicated. Schematic representations of those beams are given,

**Table 1.** Observational parameters of the main parts of the spectral patterns as indicated in Fig. 6.

| Hemisphere | Point | Opening angle | Frequency range | Magnetic latitude |
|---|---|---|---|---|
| Southern | 1S | 35° | 30–100 kHz | −50° to 0° |
| | 2S | 25° | 100–200 kHz | −40° to 0° |
| | 3S | 08° | 200–300 kHz | −15° to 0° |
| | 4S | 06° | 300–450 kHz | −12° to 0° |
| | 5S | 4.5° | 450–550 kHz | −12° to −5° |
| | 6S | 4° | 550–700 kHz | −12° to −6° |
| | 7S | 3.5° | 700–730 kHz | −10° to −6° |
| Northern | 7N | 2° | 630–700 kHz | −2° to 5° |
| | 6N | 2° | 530–580 kHz | −3° to 5° |
| | 5N | 7° | 350–480 kHz | −2° to 15° |
| | 4N | 7° | 250–350 kHz | 0° to 12° |
| | 3N | 12° | 150–250 kHz | 0° to 20° |
| | 2N | 18° | 70–150 kHz | 0° to 25° |
| | 1N | 25° | 30– 70 kHz | 0° to 30° |

respectively, in Figs. 7 and 8 for the Southern Hemisphere and Northern Hemisphere.

## 3   Discussion

We discuss hereafter the frequency-banded emission as detected by the DEMETER microsatellite. First we emphasize the beaming of such emissions and how it extended and restrained around the magnetic equatorial plane. Then the similarity and the discrepancy between DEMETER frequency-banded emission and the terrestrial kilometric radiations are addressed. This is followed by a discussion on the generation mode and the source location.

### 3.1   Beaming of the frequency-banded emission

The passage of the DEMETER satellite through the magnetic equator lead to a characterization of the frequency-banded radiation recorded in the vicinity of the magnetic equatorial plane. The capability of the DEMETER satellite leads to regularly recording such a type of emissions at low altitudes around 700 km. We have found that the radiations exhibit different spectral patterns when the frequency is smaller or bigger than 50 kHz. The satellite recorded emissions on both sides of the magnetic equator, and they appear to be more structured bands in the Northern Hemisphere. Those lasting bands indicate "stable" features in the late-evening sector at about 22 LT.

The power level distribution of the frequency-banded emission shows restrained and extended deployment around the equatorial magnetic plane. Hence the latitudinal beam is found to be about 40° when the frequency is, on average, less than 100 kHz. Above this limit and up to about 800 kHz, the latitudinal beam is decreasing and found to be

about 20°. This general picture is easily seen in Fig. 5c. However we note a clear difference in the beam when the level is less than $1\,\mu V\,m^{-1}\,Hz^{-1/2}$, as shown in Fig. 5a. Hence the frequency-banded wave radiation beam is different when combing the emission recorded in the southern and northern parts of the magnetic equatorial plane. In the southern one, half of the spectral pattern is observed, i.e. beams of −25 and 10°, on average, in the frequency bandwidths 30–100 and 100–800 kHz, respectively.

The beams of the frequency-banded events are found to depend on the satellite orbits with regard to the magnetic equatorial plane as shown in Figs. 7 and 8. Beams associated with the Southern Hemisphere events are observed in different frequency bandwidths. We may be deal with two source regions localized in the southern part of the magnetic equator but confined to two unlike regions with high and low plasma densities. On the other side of the equatorial magnetic plane, there are only branches or limbs. It is evident that emission diagrams are unlike that which may be due to the combined effects of multiple beams associated with sources localized in different regions. Beams in Figs. 7 and 8 may be considered an overlapping of single beams. Each one can be associated with one narrow-band structure as shown in Figs. 1 and 2.

Figure 9 displays the variation of the $L$ shell associated with the frequency-banded events versus the magnetic latitude of the satellite. The power level is principally found to increase in the $L$-shell range between 1 and 1.4 when the magnetic latitude of DEMETER is in between −20 and +20°. Those orbital parameters are related to the beam radiated by the source emission which crossed DEMETER trajectories.

### 3.2   Similarity and discrepancy with the terrestrial kilometric emission

Frequency-banded emission features, as investigated in this paper, allow us to address questions concerning its origin. We have found some spectral patterns which are similar to those reported in the literature in the case of the terrestrial kilometric emissions.

First, we have described changes of the spectral emissions at frequencies of about 50 kHz. Such frequencies boundaries are similar to those observed by other satellite observations, like Cluster, Geotail and IMAGE. Hence the terrestrial kilometric radiation is trapped and escaping when the frequency is, respectively, smaller and bigger than 50 kHz. The spectral features are often comparable, and the main alternations may be due to the instrumental time, frequency resolutions and also the satellite orbits with regard to the source locations. Hence Green and Boardsen (2006) show a typical sample of the kilometric continuum recorded by the RPI experiment on board IMAGE during the passage of the magnetic equatorial plane. In their Fig. 2, one can observe the presence of parallel narrow bands at frequencies above 30 kHz. Such narrow bands have a morphological similarity with those dis-

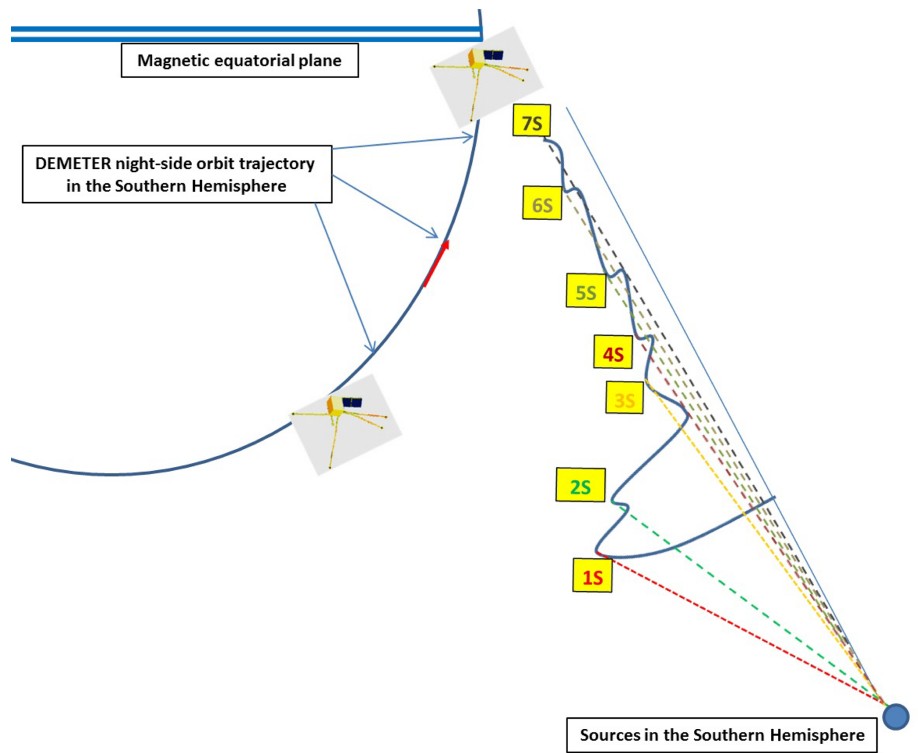

**Figure 7.** Sketch of the beams observed in the Southern Hemisphere for specific magnetic latitudes and frequencies (i.e. 1S, 2S, 3S, 4S, 5S, 6S and 7S) as listed in Table 1.

played in Fig. 6 of our paper. The AKR-X (Analyzer of Kilometric Radio emissions eXperiment) experiment on board INTERBALL-1 provided similar emissions particularly in the Southern Hemisphere at low magnetic latitudes with an $L$-shell value of about 1.2, as reported by Kuril'chik et al. (2001, 2007). Observations at fixed frequencies (100, 252, 500 and 749 kHz) allowed for the analysis of the spectral character of such emissions. Authors showed that the terrestrial kilometric radiation occurrence depends on the solar activity. Such radiation is regularly recorded during quiet solar activity. Our observations were registered in the begging of the year 2010, nearly 18 months after the minimum of solar activity, i.e. August 2008. Also the spectral pattern looks like a Christmas tree, as also reported by Green and Boardsen (2006) in their review of the kilometric continuum radiation, and it is confined to the magnetic equatorial plane.

Despite those common spectral features, several other observational aspects are different when combining the terrestrial kilometric radiation and the frequency-banded wave emission. The investigated DEMETER emission is detected at distance of about 1.1 $R_{\mathrm{E}}$, which is generally not the case of the terrestrial kilometric emission. For instance, Geotail and Cluster observations recorded radiation at more than 15 $R_{\mathrm{E}}$, as reported by Hashimoto et al. (1999) and Décréau et al. (2004), respectively. Also, the trapped or the escaping component is linked to terrestrial kilometric radiation recorded,

respectively, between the plasmasphere and the magnetosphere or outside of the magnetosphere. This radiation propagate largely in the free space in the LO (left-hand polarized ordinary) mode above the local plasma frequency linked to sources at or very near the plasmapause (Hashimoto et al., 2006). Also the gyro-frequency is found to be smaller than the trapped and the escaping frequencies, as recorded by the RPI experiment on board IMAGE (see Fig. 2 of Green and Boardsen, 2006). All those observational parameters are not similar to those reported in the case of the frequency-banded wave emission recorded by DEMETER satellite.

### 3.3 Micro-scale features of the inner part of the plasmasphere

It is clear that both radiations have common spectral features but several discrepancy observational aspects linked to the generation mechanism. However the source locations should be the plasmasphere. Hence the terrestrial kilometric radiation is linked to plasmaspheric sources with emission beams oriented towards the magnetosphere. Figures 7 and 8 showed the emission beams which interact with DEMETER orbits. The sources are localized poleward of the plasmasphere. This means that the DEMETER orbits cross the plasmaspheric hollow cones for a few dozen kilometres. Probably such restricted regions may be associated with the Z-mode waves, which are linked to the free escaping LO mode as suggested

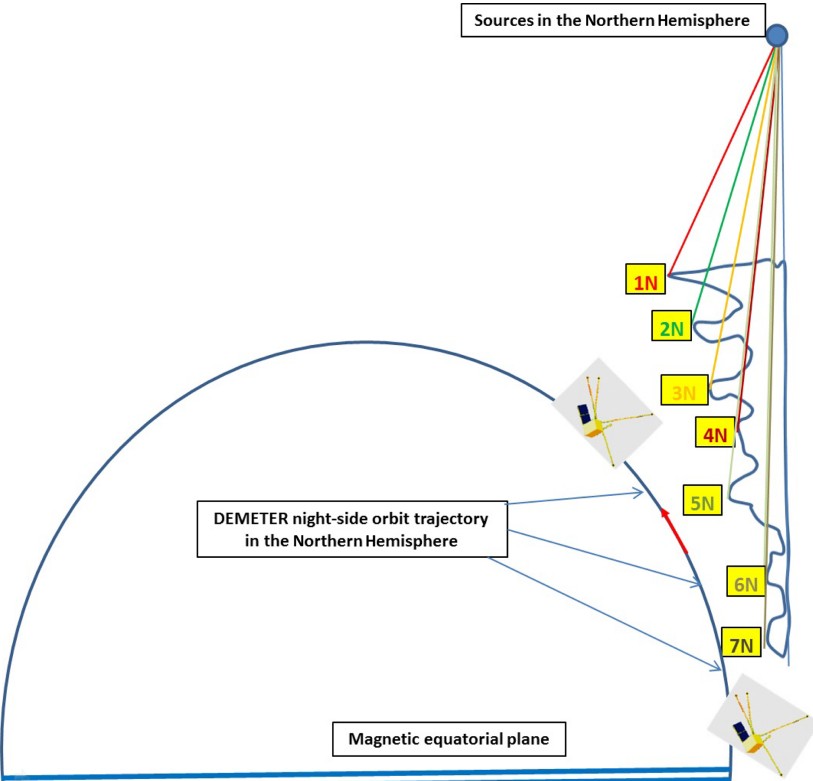

**Figure 8.** Sketch of the beams observed in the Northern Hemisphere for specific magnetic latitudes and frequencies (i.e. 1N, 2N, 3N, 4N, 5N, 6N and 7N) as listed in Table 1.

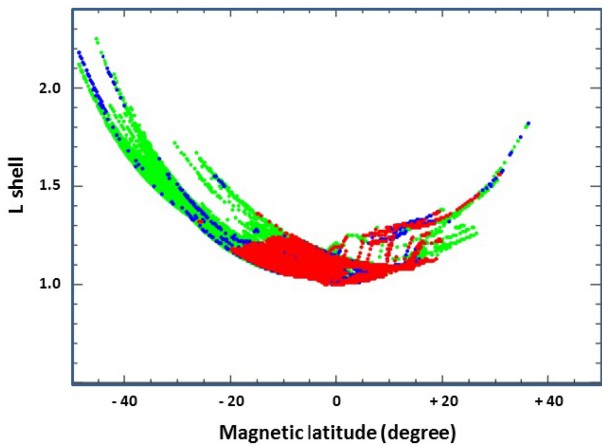

**Figure 9.** Variation of the frequency-banded wave emission versus the $L$ shell and the magnetic latitude.

by Jones (1976) in his model. In such a region, the $Z$-mode waves are considered to be trapped and later converted into the LO mode associated with the terrestrial kilometric radiation. Later on, Goertz and Strangeway (1995) derived from the Appleton–Hartree dispersion relation the whistler wave propagation in the case when the electron plasma frequency is greater CE2 than the gyro-frequency. Carpenter et al. (2003)

found a similar region where ray paths of $Z$-mode echoes from radio sounding were recorded by the IMAGE satellite in the polar regions. Also Sonwalkar et al. (2004) investigated the whistler mode echoes from radio sounding and found $f_p$ to be smaller than $f_g$ in the region poleward of the plasmapause. Green and Boardsen (2006) investigated and reported on the linear-mode-conversion theory based on the model of Jones. Authors showed profiles of plasmaspheric plasma frequency, taking into consideration the $Z$-mode frequency and the equatorial gyro-frequency. Regions of a sharp plasma gradient are found and shown in Fig. 5 of their paper.

We estimate the relationship between the $Z$-mode frequency ($f_z$), the plasma frequency ($f_p$) and the gyro-frequency ($f_g$) using the following formula: $f_z = (f_g/2)[-1 + (1 + 4(f_p/f_g)^2)^{1/2}]$ (Carpenter et al., 2003). Figure 10 displays the variation of the three frequencies (i.e. $f_z$, $f_p$ and $f_g$) versus the geocentric distance. The trapping region is localized between the lower and the higher $f_z$ (green in Fig. 10), mainly between 1 and 100 kHz and extended up to 700 kHz. The plasma frequency follows the trapping region, starting at about 10 and going up to 800 kHz. The gyro-frequency appears at higher frequencies, i.e. above 800 kHz. Those features are comparable to previous investigations, e.g. Gurnett et al. (1983) and Carpenter et al. (2003), in polar regions.

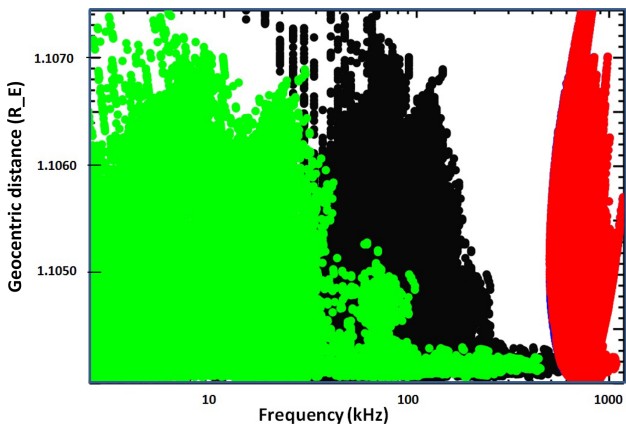

**Figure 10.** Variations of the frequency-banded emission versus the geocentric distance expressed in $R_E$. The green, black and red are associated, respectively, with $f_z$, $f_p$ and $f_g$ frequencies.

It is important to note that the frequency-banded emissions may be due to the interaction and/or the co-existence of electrostatic and whistler waves. Hence Bell and Ngo (1990) considered theoretically the generation, in the case of the density cavity or gradient, of whistler waves by lower-hybrid waves. Such a co-existence of both waves has been investigated by An et al. (2017) using Darwin particle-in-cell (PIC) simulation. Such models have been recently invoked by Vartanyan et al. (2016) and Li et al. (2017) in the investigation, respectively, of the generation of whistler waves and chorus wave modulation observed on board DEMETER and the Van Allen Probes.

## 4  Conclusion

We have investigated the frequency-banded wave radiation recorded by the ICE experiment on board DEMETER. DEMETER orbits allow us to regularly record this radiation where, in the optimal case, about 13 events are daily registered. The power level is found in the interval between $10^{-3}$ and $10^{+4}\,\mu V\,m^{-1}\,Hz^{-1/2}$. The spectral analysis leads to finding a "tree spectral shape", which is the traces of the beaming of the frequency-banded wave radiation. We have shown that those beams are not similar and depend on the emission frequency and the magnetic latitude. The DEMETER frequency-banded emission can be comparable to the well-know terrestrial kilometric radiation. However several other observational aspects are different when combining both emissions, in particular the generation modes. We suggest that the DEMETER frequency-banded emissions are linked to a $Z$-mode micro-scale region. This trapping $Z$-mode region can only be detected between Earth's ionosphere and the plasmasphere. The hollow cones of these frequency-banded wave emissions are crossed by the DEMETER orbits at altitudes lower than 700 km. Probably the source regions of

the DEMETER frequency-banded emission should be the plasmasphere, like the terrestrial kilometric radiation. IMAGE investigations have reported about the time evolution of the plasmasphere in particular on the pre-midnight sector (Sandel et al., 2003) and the presence of density structures (Darrouzet et al., 2009) at smaller scales. We may consider that DEMETER orbits allow for investigating the inner part of the plasmasphere when other missions (i.e. Geotail, IMAGE and INTERBALL) lead to studying the outer part of the plasmasphere.

*Data availability.* Instrument Champ Électrique (ICE) data used in this study are available from the DEMETER Data Server (http:// demeter.cnrs-orleans.fr/[TS1], last access: 13 May 2019).

*Author contributions.* MYB carried out the analysis and wrote the paper. PHMG, VD and HL helped with the interpretation of the data. All authors contributed to the discussion.

*Competing interests.* The authors declare that they have no conflict of interest.

*Acknowledgements.* Authors thank the Centre National d'Etudes Spatiales (CNES) and Jean-Jacques Berthelier (PI of the ICE instrument) for the use of the data. We acknowledge support by the Austrian Academy of Sciences and the Russian Academy of Sciences. The authors are grateful to the anonymous reviewers, whose comments helped them to improve the article.

*Financial support.* This research has been supported by the Austrian Academy of Sciences and the Russian Academy of Sciences[TS2].

*Review statement.* This paper was edited by Yoshizumi Miyoshi and reviewed by two anonymous referees.

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

**Remarks from the language copy-editor**

CE1    There was a mistake on our side in which Figures was abbreviated to Figs. at the beginning of the sentence (where it must be unabbreviated). I have corrected this mistake. No further action is needed.

CE2    Please give an explanation of why this needs to be changed. We need to inform the editor about the exact reason for the change (it could be as simple as a typo).

**Remarks from the typesetter**

TS1    Data sets should be cited as individual contributions. Therefore, if applicable, I would kindly ask you to provide a reference mentioning authors, title, URL and last-access date.

TS2    Information about the financial support should be kept in this section (unless it does not directly apply to this study).