# Peer review of "Low altitude frequency banded equatorial emissions observed below"

_Annales Geophysicae, 2019_

## Referee Comment (RC1) · Anonymous Referee #1 · 12 Apr 2019

Review of 'Terrestrial kilometric radiation observed on pre-midnight side of the Earth at 1-2 L-Shell' by Boudjada for publication in the Journal Ann. Geophys..

The authors state that kilometric continuum (KC) is observed by the DEMETER spacecraft, which is at an altitude of ~700 km. The scatter plot in Figure 4 shows the bulk of the emissions occurring below 100 kHz. At 100 kHz the plasma density is ~120 cm$^{-3}$. Since KC is a free space mode radiation its frequency (f) must be above $F_{PE}$ or $F_R$ [eg. Shaw and Gurnett, 1980, Kennel et al., 1987]. I don't know the DEMETER location for Figure 1 so used 2 magnetic field strengths in the table below.

| $N_E$(cm$^{-3}$) | B(nT) | $F_{PE}$ (kHz) | $F_{CE}$ (kHz) | $F_R$ (kHz) | $F_L$ (kHz) |
|---|---|---|---|---|---|
| 10000 | 30000 | 900 | 839 | 1413 | 573 |
| 1000 | 30000 | 284 | 839 | 927 | 87 |
| 100 | 30000 | 90 | 839 | 849 | 9 |
| 10000 | 20000 | 900 | 559 | 1222 | 662 |
| 1000 | 20000 | 284 | 559 | 679 | 119 |
| 100 | 20000 | 90 | 559 | 573 | 14 |
| The cutoffs $F_R$ is f where R=0, $F_L$ is f where L=0; R&L are from Stix [1962] | | | | | |
| $F_{PE}$ -plasma frequency, $F_{CE}$ -electron cyclotron frequency. | | | | | |

The authors suggest that the emissions observed by DEMETER could be related to plasmaspheric plumes. For one to believe these emissions observed by DEMETER are free-space KC, a plasmaspheric drainage plume/channel would have to extend down to ~700 km altitude with minimum densities of say ~ 100 cm$^{-3}$ or less at that altitude; an altitude where ~ $10^4$ to $4 \times 10^4$ cm$^{-3}$ is common near the equator. Looking at Figures 2 and 3 of Chen et al., [2018] $N_E$ dropping below $10^4$ cm$^{-3}$ on the nightside or dayside around the equator is rare. So, if DEMETER is frequently seeing this emission around the equator then I don't believe these emissions are free-space.

Z-mode radiation occurs at f between $F_L$ and $F_{UHR}$ (upper hybrid resonance f) the authors need to check if this radiation could be Z-mode.

MAJOR COMMENT:

A key issue must be resolved, before the reviewer can accept this paper for publication.

Since this is a major claim that DEMETER detects KC, a detailed event analysis should be given in the paper that demonstrates the radiation is free space mode.

You have not convinced me that your type 1&2 events are free space radiation.

I suggest that you show one of more example spectrograms of events with the frequency of the &$F_{PE}$& &$F_R$& $F_L$ & $F_{CE}$ lines overlaid on the spectrogram. This will give the reader a feeling of whether or not the radiation is free-space.

If ISL Langmuir probe paper Lebreton et al. [Planetary and Space Science 54 (2006) 472–486] data is not available for any of your events, you could try IAP or infer the plasma density from the E/B ratio using data below 17.4 kHz, assuming you can identify the wave mode and the E&B measurements are reliable. 100 per cc might be the threshold of ISL so f(l=0) & fpe & fuhr would be a upper limit.

and

If you can correlate some of your events with KC events observed by GEOTAIL PWI. The GEOTIAL PWI 24 hr survey plots are located at http://space.rish.kyoto-u.ac.jp/gtlpwi/.  I see no KC in the GEOTAIL PWI spectrograms for the 2 days given in your paper, of course GEOTAIL could be at the wrong LT or the KC generated at low RE does not always escape the plasmasphere/ionosphere.

On your spectrogram plot please indicate where the IGRF (or similar) model field aligned magnetic minimum crossing occurs. If centered about the Type 2 emissions then this would cast doubt in my opinion about the emissions being KC.

No discussion of the interpretation of the harmonics of type 2 is given. Looking at the type 2 in Figure 1, the spacing between the harmonics is ~25 kHz.  Using a simple dipole and standard continuum emission model this places the equatorial source at about ~3.2 RE, with a sharp plasma gradient, with Ne extending up to at least (fpe=600 kHz) ~4500 cm$^{-3}$ at that location.  Fpe at 600 kHz at ~3.2 RE is at the upper range of observed plasmaspheric plasma frequencies at ~3.2 RE, further casting doubt in my opinion.  It is not clear to me if harmonic spacing of ~25 kHz can be explained in terms of local plasma conditions and/or non-linear processes.

Because $F_{CE}$ is large I would like to see at least one the spectrogram of the entire ICE frequency range out to 3.25 Mega-Hz.

Have you to tried to correlate Type 2 with the particle measurements (IAP, ISL, IDP)?

It's important to understand these emissions. Have you searched for other explanations for these emissions? Could this be an example of instrumental spherical probe pre-amp oscillations due to localize plasma conditions?

OTHER COMMENTS:

Free-space or Z-mode emission in an equatorial plasma bubbles might be another possibility instead of drainage plumes. Equatorial bubbles are observed by DMSP on about 1 out 8 orbits [Huang et al., JGR 2001] whether the internal density of bubbles can be low enough to accommodate the DEMETER observations is not clear.

Line 28. 'We use a manually technique which consists to follow and to save with the PC-computer mouse'. Instead of manual selection did you try an automated selection method. Looking at Figure 1, it seems like automated selection followed by visual inspection of those selections might save one time, this would allow you to scan a larger time interval.

You don't give enough information about your survey.
Start/stop dates that your survey covered.
We looked a X nightside equatorial crossings finding Y events?
We looked a X dayside equatorial crossings finding Y events?
Where these emissions not observed before 2010? If so why, or did you not look before 2010?

Figure 1 is not of publication quality. I would also include a dayside example, maybe 2 dayside and 2 nightside examples, with better annotation as described earlier in the review.

Figure 2
Lack of clarity in how the histogram is computed: for example, looking at Figure 1 at 14:01:40 are all harmonics summed in a given latitude bin?
I would make a weighted histogram instead, summing the weights in each latitude bin. For example if you selected 800 points in Figure 1, then I would weight each selection by 1/800 for that equatorial crossing.

Why not split histogram into day/night?

Scatter plot of power versus frequency might be revealing.

An annotated spectrum at the center of Type 2 would also be helpful.

power level (defined as square root of the power spectral density)

Figure 3: include legend of what the 3 colors correspond too, don't just say it in the text, this makes it hard for the reader.

Figure 4: include legend of what the 3 colors correspond, don't just say it in the text. Why not split into day/night?

A scatter plot for Type 2 of the frequency spacing of harmonics versus frequency of harmonic might be revealing. From standard theory this can be used to estimate the fc/fp ratio at the source under the assumption that the density gradients are sharp.

References

Chen, C. Y., Liu, T. J. Y., Lee, I. T., Rothkaehl, H., Przepiorka, D., Chang, L. C., et al. (2018). The midlatitude trough and the plasmapause in the nighttime ionosphere simultaneously observed by DEMETER during 2006–2009. Journal of Geophysical Research: Space Physics, 123, 5917–5932. https://doi.org/10.1029/2017JA024840

Huang, C. Y., Burke, W. J., Machuzak, J. S., Gentile, L. C., and Sultan, P. J. ( 2001), DMSP observations of equatorial plasma bubbles in the topside ionosphere near solar maximum, *J. Geophys. Res.*, 106( A5), 8131– 8142, doi:10.1029/2000JA000319.

Kennel, C. F., Chen, R. F., Moses, S. L., Kurth, W. S., Coroniti, F. V., Scarf, F. L., and Chen, F. F.( 1987), *Z* mode radiation in Jupiter's magnetosphere, *J. Geophys. Res.*,  92( A9),  9978– 9996, doi:10.1029/JA092iA09p09978.

Shaw, R. R., and Gurnett, D. A. (1980), A test of two theories for the low-frequency cutoffs of nonthermal continuum radiation, J. Geophys. Res., 85( A9), 4571– 4576, doi:10.1029/JA085iA09p04571.

---

## Referee Comment (RC2) · Anonymous Referee #2 · 27 Apr 2019

Referee Comments

This paper reports observations of LF waves which might be generated at the plasmapause and propagate to the low-altitude equatorial region. The observation results are interesting and raised some important problems of propagation characteristics of the LF waves in the plasmasphere.

I have some comments on the paper.

1. In the manuscript, the authors do not mention an important point, that is, the observed LF waves by DEMETER are "whistler mode waves".
I attached a diagram (drawn by myself) which shows the characteristic wave-mode in the frequency range of 10kHz-2MHz in the altitude range of 100-10000km. It is obvious that the ICE/DEMETER instrument can detect only the whistler mode waves within the observation frequency range up to 3.5 MHz at the altitude of DEMETER (about 700km).

[Figure]

Thus, the observed waves by DEMETER are not free-space mode but R-X mode of whistler waves trapped in the plasmasphere. When the authors consider the LF wave propagation from the source region near the plasmapause to the low-altitude equatorial region, they should take into account the propagation characteristic of whistler mode waves with respect to the magnetic field.

2. I recommend considering the mode conversion of original radiations, which are probably Z-mode or upper hybrid mode radiation generated around the plasmapause, to whistler mode waves. And the observed kilometric radiation at DEMETER can be whistler mode wave.

However, in this case, it becomes difficult and needs some suitable idea to interpret the "beam pattern" derived from the authors' study, because the whistler mode wave tends to propagate along the magnetic field. This is the interesting point that authors raised.

3. In the text, the authors say that the type1 is trapped component and the type2 is escaping component. What is the reason?

Usually, the term of trapping/escaping is used in the case of free-space mode propagation in the magnetosphere.

4. The authors distinguished two varieties of emission:

"Type 1 appears as a narrow continuum with an instantaneous bandwidth of about 2 kHz at frequencies less than 50 kHz, and displays negative and positive frequency drifts when the satellite is approaching or leaving the equatorial plane, respectively. Its frequency drift rate is weak and in the order of 0.2 kHz/s. Type 2 is composed of parallel narrow-bands in a frequency above 50 kHz and up to 800 kHz."

I agree the presence of Type1 and Type 2 radiations, but do not agree to use mixed data of Type1 and Type2 in the analysis of figures 2-6.

I suggest that they should be separately analyzed, because the different characteristics of type1 and 2 suggest the different source mechanism and/or different propagation pass.

5. The authors found structured emissions in the LF waves, and classified into two categories:

"In the northern hemisphere, five components in the frequency ranges of few kHz - 50 kHz, 70 kHz - 130 kHz, 170 kHz - 250 kHz, 280 kHz -340 kHz and 380 kHz- 420 kHz.

In the southern hemisphere, four components in the frequency bands: of   200 kHz - 320 kHz, 320 kHz - 450 kHz, 450 kHz - 570 kHz, and 570 kHz - 670 kHz."

The reader will imagine that these bands are showing the higher harmonic relation. In fact, as shown in Figure 1, an individual event shows fine harmonics. And, one can easily infer the fundamental frequency from the harmonic relation, and then can suppose the source altitude of the emission assuming the distribution of gyrofrequency and plasma frequency. I suggest to add discussion on this matter in the text.

6. minor comments

*p1, line17    seventeens should be seventies.

*p2, line 13    plasmasphere should be magnetosphere.

*Fig 6          vertical axis is wrong.

* Unpublished paper should not be included in References.

---

## Author Comment (AC2) · 24 Jun 2019

Dear Editor,

Please find as a zip-supplement file the replay to the referee comments and the up-graded version of the paper.

Best regards,

Boudjada Mohammed.

Please also note the supplement to this comment:

[Figure]

https://www.ann-geophys-discuss.net/angeo-2019-48/angeo-2019-48-AC2-supplement.zip

---

## Author Response (AR1)

**Response to the Referees**

**Kilometric wave emission observed on pre-midnight side in the vicinity of the Earth's magnetic equatorial plane at 1-2 L-Shell**

First of all, we thank both reviewers for their constructive comments and suggestions. In the title, we have written 'Kilometric wave emission' instead of 'terrestrial kilometric radiation', and indicated where DEMETER detected such radiation. In the upgraded version we have re-considered and re-written essentially the Section 3 taking into consideration the reviewer comments. It is evident that the confusions between DEMETER kilometric wave emission and the terrestrial kilometric radiation are due to three main reasons: spectral beam, i.e. 'Christ tree', are similar, the parallel bands and the beaming around the magnetic equatorial plane. In the discussions Section, i.e. Section 3, we have summarized the main results and insisted on one side, on the spectral features of the kilometric wave emissions and on the other side on the similarities and the discrepancies between the DEMETER kilometric emission and the terrestrial kilometric radiation. The Z-mode is considered as a generation mechanism candidate. Our responses to referee are listed below.

**Response to Referee #1**

*Reviewer1: The authors state that kilometric continuum (KC) is observed by the DEMETER spacecraft, which is at an altitude of ~700 km. The scatter plot in Figure 4 shows the bulk of the emissions occurring below 100 kHz. At 100 kHz the plasma density is ~120 cm-3. Since KC is a free space mode radiation its frequency (f) must be above FPE or FR [eg. Shaw and Gurnett, 1980, Kennel et al., 1987]. I don't know the DEMETER location for Figure 1 so used 2 magnetic field strengths in the table below.*

| $N_E$(cm$^{-3}$) | B(nT) | $F_{PE}$ (kHz) | $F_{CE}$ (kHz) | $F_R$ (kHz) | $F_L$ (kHz) |
|---|---|---|---|---|---|
| 10000 | 30000 | 900 | 839 | 1413 | 573 |
| 1000 | 30000 | 284 | 839 | 927 | 87 |
| 100 | 30000 | 90 | 839 | 849 | 9 |
| 10000 | 20000 | 900 | 559 | 1222 | 662 |
| 1000 | 20000 | 284 | 559 | 679 | 119 |
| 100 | 20000 | 90 | 559 | 573 | 14 |
| The cutoffs $F_R$ is f where R=0, $F_L$ is f where L=0; R&L are from Stix [1962] | | | | | |
| $F_{PE}$ -plasma frequency, $F_{CE}$ -electron cyclotron frequency. | | | | | |

*The authors suggest that the emissions observed by DEMETER could be related to plasmaspheric plumes. For one to believe these emissions observed by DEMETER are free-space KC, a plasmaspheric drainage plume/channel would have to extend down to ~700 km altitude with minimum densities of say ~ 100 cm-3 or less at that altitude; an altitude where ~ 104 to 4x104 cm-3 is common near the equator. Looking at Figures 2 and 3 of Chen et al., [2018] NE dropping below 104 cm-3 on the nightside or dayside around the equator is rare. So, if DEMETER is frequently seeing this emission around the equator then I don't believe these emissions are free-space. Z-mode radiation occurs at f between FL and FUHR (upper hybrid resonance f) the authors need to check if this radiation could be Z-mode. A key issue must be resolved, before the reviewer can accept this paper for publication.*

**MAJOR COMMENTS**

*R1_A:  A key issue must be resolved, before the reviewer can accept this paper for publication. Since this is a major claim that DEMETER detects KC, a detailed event analysis should be given in the paper that demonstrates the radiation is free space mode. You have not convinced me that your type 1&2 events are free space radiation. I suggest that you show one of more example spectrograms of events with the frequency of the & FPE & & FR & FL & FCE lines overlaid on the spectrogram. This will give the reader a feeling of whether or not the radiation is free-space. If ISL Langmuir probe paper Lebreton et al. [Planetary and Space Science 54 (2006) 472–486] data is not available for any of your events, you could try IAP or infer the plasma density from the E/B ratio using data below 17.4 kHz, assuming you can identify the wave mode and the E&B measurements are reliable. 100 per cc might be the threshold of ISL so f(l=0) & fpe & fuhr would be a upper limit.*

A1_A: As suggested by the referee, we have checked and found that the considered kilometric wave emissions can't a free space radiation. In all events the gyrofrequency (FCE) is above the frequency associated to the LF kilometric emission. In Fig.1 or Fig.2 the gyro-frequency is indicated.

*R1_B:  If you can correlate some of your events with KC events observed by GEOTAIL PWI. The GEOTIAL PWI 24 hr survey plots are located at http://space.rish.kyotou.ac.jp/gtlpwi/. I see no KC in the GEOTAIL PWI spectrograms for the 2 days given in your paper, of course GEOTAIL could be at the wrong LT or the KC generated at low RE does not always escape the plasmasphere/ionosphere.*

A1_B: GEOTAIL orbits were mainly far, at least 10 RE, and on the day side. We have checked the PWI dynamic spectra for the investigated period and did not find comparable spectral features as recorded by DEMETER. This may be due to the distance of the satellite which is bigger than 5 RE.

*R1_C: On your spectrogram plot please indicate where the IGRF (or similar) model field aligned magnetic minimum crossing occurs. If centered about the Type 2 emissions then this would cast doubt in my opinion about the emissions being KC.*

A1_C: As suggested by the referee, we have indicated in Fig.1 and Fig.2 the gyrofrequency.

*R1_D: No discussion of the interpretation of the harmonics of type 2 is given. Looking at the type 2 in Figure 1, the spacing between the harmonics is ~25 kHz. Using a simple dipole and standard continuum emission model this places the equatorial source at about ~3.2 RE, with a sharp plasma gradient, with Ne extending up to at least (fpe=600 kHz) ~4500 cm-3 at that location. Fpe at 600 kHz at ~3.2 RE is at the upper range of observed plasmaspheric plasma frequencies at ~3.2 RE, further casting doubt in my opinion.*

A1_D: We totally agree concerning the harmonics of type 2. The frequency spacing of 25 kHz of Figure 1, recorded mainly in the southern part of magnetic equator, is variable from one event to another. This structured component disappears when we consider all events, i.e. as can see in the first panel of Fig.4. However in the northern magnetic equatorial plane, the frequency interval is, on average, about 150 kHz at magnetic latitude 20°N.

*R1_E: It is not clear to me if harmonic spacing of ~25 kHz can be explained in terms of local plasma conditions and/or non-linear processes. Because FCE is large I would like to see at least one the spectrogram of the entire ICE frequency range out to 3.25 Mega-Hz.*

A1_E: We have added in Fig.1 and Fig.2 an overview (i.e. total frequency range) and a zoomed part (i.e. from few kHz up to about 900 kHz) for the two examples.

*R1_F: Have you to tried to correlate Type 2 with the particle measurements (IAP, ISL, IDP)? It's important to understand these emissions? Have you searched for other explanations for these emissions? Could this be an example of instrumental spherical probe pre-amp oscillations due to localize plasma conditions?*

A1_F: In reality, we did not check the particle measurement on aboard DEMETER. This may be done in further investigation of this work. Of course, we did not try to search for other explanations since we thought that we deal with terrestrial kilometric radiation particularly because of the spectral beam. Concerning the instrumental influence, Bertherlier (PI of ICE experiment) did not address such instrumental effects in his paper (i.e. Berthelier et al., 2006). The frequency bandwidth is not constant but variable from one event to another.

**OTHER COMMENT**

*R1_G: Free-space or Z-mode emission in an equatorial plasma bubbles might be another possibility instead of drainage plumes. Equatorial bubbles are observed by DMSP on about 1 out 8 orbits [Huang et al., JGR 2001] whether the internal density of bubbles can be low enough to accommodate the DEMETER observations is not clear.*

A1_G: We though to the drainage plumes because of the development of the emission beam in the case of the terrestrial kilometric radiation. In the upgraded version, we only refer to the source location as reported by Green & Boardsen (2006) and avoid the confusion between both kilometric emissions.

*R1_H: Line 28. 'We use a manually technique which consists to follow and to save with the PC-computer mouse'. Instead of manual selection did you try an automated selection method. Looking at Figure 1, it seems like automated selection followed by visual inspection of those selections might save one time, this would allow you to scan a larger time interval.*

A1_H: The manual technique has been adapted because of: (a) the weak intensities of kilometric wave emission when compared to AKR and also to the instrumental noise level, (b) the phenomenological aspects of this emission where we have attempted to classify/distinguish other spectral components observed at mid-latitude and sub-auroral regions and (c) the presence of LF transmitters which are overlapping the investigated kilometric wave emission.

*R1_I: You don't give enough information about your survey. Start/stop dates that your survey covered. We looked a X nightside equatorial crossings finding Y events? We looked a X dayside equatorial crossings finding Y events?*

A1_I: In the paper, we have indicated the probability of occurrence of such kilometric wave emissions observed only on the night-side of the Earth. We found that the crossing of the magnetic equatorial plane by DEMETER is usually followed by the detection of kilometric wave emissions, as displayed in Fig.4. The difference from one orbit to another is the intensity level of the emission and also the frequency bandwidth which is found in the range between few kHz and 800 kHz.

*R1_J: Where these emissions not observed before 2010? If so why, or did you not look before 2010?*

A1_J: We started in the beginning of 2010 because of the low solar activity. We followed the work of Kuril'chik et al. (2001) who reported that the terrestrial kilometric radiation is regularly observed during quiet solar activity.

*R1_K: Figure 1 is not of publication quality. I would also include a dayside example, maybe 2 dayside and 2 nightside examples, with better annotation as described earlier in the review.*

A1_K: As suggested by the referee, we have considered two examples with better annotations. We have no dayside events.

*R1_L:  Figure 2 Lack of clarity in how the histogram is computed: for example, looking at Figure 1 at 14:01:40 are all harmonics summed in a given latitude bin? I would make a weighted histogram instead, summing the weights in each latitude bin. For example if you selected 800 points in Figure 1, then I would weight each selection by 1/800 for that equatorial crossing.*

A1_L: We attempt in this paper to provide a global view of the occurrence of all events. The use of manually method is not adequate for 'weighting' the bin in latitude and longitude. For the data processing, we did not use conditions on: (a) the time and frequency spacing between two points, and (b) on the intensity level. We have emphasized on the spectral pattern of the emissions like the fluctuation in time and frequency, and the variable frequency bandwidth.

*R1_M: Why not split histogram into day/night? Scatter plot of power versus frequency might be revealing. An annotated spectrum at the center of Type 2 would also be helpful. Power level (defined as square root of the power spectral density)*

A1_M: We observed the kilometric wave emission only when DEMETER was on the night-side. Fig.4, 5, 6, and 7 displayed the dependence of the emission frequency on the power levels. We have considered three intensity levels associated to the three maxima derived from the second panel of Fig.3. The power level has been corrected and expressed as $\mu V\ m^{-1}Hz^{-1/2}$.

*R1_N:  Figure 3: include legend of what the 3 colors correspond too, don't just say it in the text, this makes it hard for the reader.*

A1_N: The colors associated to the power levels are corrected and indicated in the new legend of Fig.3.

*R1_O:  Figure 4: include legend of what the 3 colors correspond, don't just say it in the text. Why not split into day/night?*

A1_O: We also precise in the new legend of Fig.4 the corresponding power levels.

*R1_P:  A scatter plot for Type 2 of the frequency spacing of harmonics versus frequency of harmonic might be revealing. From standard theory this can be used to estimate the fc/fp ratio at the source under the assumption that the density gradients are sharp.*

A1_P: We have previously indicated that the frequency bandwidth is variable. The assumption about the density gradients have been discussed in the upgraded version if the observed emission is the terrestrial kilometric radiation. The fc/fp ratio may be applied if the plasma frequency is higher than the gyrofrequency frequency. It is not the case in this study.

First of all, we thank both reviewers for their constructive comments and suggestions. In the title, we have written 'Kilometric wave emission' instead of 'terrestrial kilometric radiation', and indicated where DEMETER detected such radiation. In the upgraded version we have re-considered and re-written essentially the Section 3 taking into consideration the reviewer comments. It is evident that the confusions between DEMETER kilometric wave emission and the terrestrial kilometric radiation are due to three main reasons: spectral beam, i.e. 'Christ tree', are similar, the parallel bands and the beaming around the magnetic equatorial plane. In the discussions Section, i.e. Section 3, we have summarized the main results and insisted on one side, on the spectral features of the kilometric wave emissions and on the other side on the similarities and the discrepancies between the DEMETER kilometric emission and the terrestrial kilometric radiation. The Z-mode is considered as a generation mechanism candidate. Our responses to referee are listed below.

**Response to Referee #2**

*Reviewer2: This paper reports observation of LF waves which might be generated at the plasmapause and propagate to the low-altitude equatorial region. The observation results are interesting and raised some important problems of propagation characteristics of the LF waves in the plasmasphere. I have some comments on the paper.*

[Figure]

*R2_A: 1. In the manuscript, the authors do not mention an important point, that is, the observed LF waves by DEMETER are "whistler mode waves". I attached a diagram (drawn by myself) which shows the characteristic wave-mode in the frequency range of 10 kHz-2 MHz in the altitude range of 100-10000km. It is obvious that the ICE/DEMETER instrument can detect only the whistler mode waves within the observation frequency range up to 3.5 MHz at the altitude of DEMETER (about 700km). Thus, the observed waves by DEMETER are not free-space mode but R-X mode of whistler waves trapped in the plasmasphere. When the authors consider the LF wave propagation from the*

*source region near the plasmapause to the low-altitude equatorial region, they should take into account the propagation characteristic of whistler mode waves with respect to the magnetic field.*

A2_A: We agree with the referee suggestion concerning the whistler mode waves as a physical process at the origin of this emission. The attached diagram, provided by the referee, leads to explain the observed frequencies and their corresponding altitudes. In the upgraded version, we report about new references concerning whistler mode waves and particularly the Z-mode which may be observed in the vicinity of the magnetic equatorial plane.

*2_B: 2. I recommend considering the mode conversion of original radiations, which are probably Z-mode or upper hybrid mode radiation generated around the plasmapause, to whistler mode waves. And the observed kilometric radiation at DEMETER can be whistler mode wave. However, in this case, it becomes difficult and needs some suitable idea to interpret the "beam pattern" derived from the authors' study, because the whistler mode wave tends to propagate along the magnetic field. This is the interesting point that authors raised.*

A2_B: We did not change too much in the content of the Section 2 when it is compared to the previous one. We have mainly derived the observational parameters particularly the variation of the power level versus the frequency and the magnetic latitude. In Section 3, we have attempted to explain the similarities and the discrepancies between the DEMETER kilometric emission and the well-known terrestrial kilometric radiations.

*R2_C: 3. In the text, the authors say that the type1 is trapped component and the type2 is escaping component. What is the reason? Usually, the term of trapping/escaping is used in the case of free-space mode propagation in the magnetosphere.*

A2_C: It is clear from the dynamic spectrum of kilometric wave emission that we deal with two components. The spectral shapes, as shown in Fig.5, are found to be similar to those associated to the terrestrial kilometric emission. For this reason we have considered the trapped (Type 1) and escaping (Type 2) emission taking into consideration the frequency boundaries around 50 kHz. Such spectral patterns are addressed and discussed in Section 3.

*R2_D: 4. The authors distinguished two varieties of emission: "Type 1 appears as a narrow continuum with an instantaneous bandwidth of about 2 kHz at frequencies less than 50 kHz, and displays negative and positive frequency drifts when the satellite is approaching or leaving the equatorial plane, respectively. Its frequency drift rate is weak and in the order of 0.2 kHz/s. Type 2 is composed of parallel narrow-bands in a frequency above 50 kHz and up to 800 kHz."*
*I agree the presence of Type1 and Type 2 radiations, but do not agree to use mixed data of Type1 and Type2 in the analysis of figures 2-6. I suggest that they should be separately analyzed, because the different characteristics of type1 and 2 suggest the different source mechanism and/or different propagation pass.*

A2_D: In the upgraded version, we have described both types of emissions. However, we don't consider them as trapped and escaping emissions. We have attempted to insist on the source regions of both components as discussed in Section 3. Further surveys of ICE observations may allow a better characterization of each component by considering a longer period of investigations, at least one year.

*R2_E: 5. The authors found structured emissions in the LF waves, and classified into two categories: "In the northern hemisphere, five components in the frequency ranges of few kHz - 50 kHz, 70 kHz - 130 kHz, 170 kHz - 250 kHz, 280 kHz -340 kHz and 380 kHz- 420 kHz.*
*In the southern hemisphere, four components in the frequency bands: of 200 kHz - 320 kHz, 320 kHz - 450 kHz, 450 kHz - 570 kHz, and 570 kHz - 670 kHz." The reader will imagine that these bands are showing the higher harmonic relation. In fact, as shown in Figure 1, an individual event shows fine harmonics. And, one can easily infer the fundamental frequency from the harmonic relation, and then can suppose the source altitude of the emission assuming the distribution of gyrofrequency and plasma frequency. I suggest to add discussion on this matter in the text.*

A2_E: In Section 3, we have suggested the probable source regions of the kilometric wave emissions as a micro-scale region in the inner plasmasphere. In the new Fig.9, we have attempted to display how the Z-mode frequency is delimiting the source altitude.

*R2_F: 6. minor comments: \*p1, line17 seventeens should be seventies, \*p2, line 13 plasmasphere should be magnetosphere.\*Fig 6 vertical axis is wrong. \* Unpublished paper should not be included in References.*

A2_F: Minor comments are considered in the upgraded version. Unpublished references (i.e. Boudjada et al., EGU09 & Boudjada et al., EGU14) have been deleted from the text and the reference list.

[revised manuscript text omitted]

---

## Referee Report (RR1)

There are many problems and confusing descriptions in the Sections 3 and 4.

(1) Page 7, line 54

"The power level is principally found to increase between 1 and 1.4 L-shell when the magnetic latitude of DEMETER is in between -20◦ and +20◦. The source locations of kilometric wave radiation seem to be confined to a narrow L-shell region."

This means that the source region of DEMETER kilometric wave emission is located in the altitude range from about 6000 to 9000 km above the equatorial upper ionosphere (-20 to +20 MLAT).

However, in page 9, line 49

"the beaming of the DEMETER kilometric wave emissions is towards the Earth's ionosphere with sources localized in the plasmasphere."

These two sentences are inconsistent.

(2) Page 8, line 48-

."the parallel narrow bands as displayed inFig.5 are mainly associated to the escaping continuum in the case of the terrestrial kilometric emission."

What is the meaning of "associated to the escaping ····". Are they same category waves in the dispersion relation or simply morphological similarity of their spectra?

(3) Page 8, line 58

"AKR-X/INTERBALL-1experiment provided similar emissions particularly in the southern hemisphere at low magnetic latitude and at L-Shell of about 1.2 (Kuril'chik et al., 2001)."

"L-Shell of about 1.2" is wrong, and should be "R/RE of about 1.2".

  The INTERBALL did not provide the data at low L-Shell region (low altitude and low latitude), but at low R/RE region (low altitude and high latitude).

(4) Again page 9, line 49

"the beaming of the DEMETER kilometric wave emissions is towards the Earth's ionosphere with sources localized in the plasmasphere."

  If so, how the waves can propagate to the low altitude equatorial ionosphere through the dense plasma region? What is the propagation mode?

(5) Page 9, line 51

"It is important to note that the DEME-TER kilometric events belongs to a very limited regions in range 1-2 L-Shell (as shown in Fig.7) with distances between 1.1040 RE and 1.1070 RE. This means that the DEMETER orbits is crossing the plasmaspheric hollow cones on few dozen of kilometer."

  What is the "hollow cone"?

There is no explanation. Physical explanation is necessary.

It should be strongly related to the wage generation process.

(6) Figure 9

I suppose that this figure is based on the orbital plasma data from equatorial to polar region where fp, fg, and fz changes greatly . If so, this figure is not applicable for your discussion, because the wave propagation mode to be considered here is restricted to that of the low latitude inner plasma sphere.

(7) Page 9, line 56

"Probably such restricted regions may be associated to the Z-mode waves which are linked to the free escaping L-O mode as suggested by Jones (1976) in his model. In such region the Z-mode waves are considered to be trapped and later converted into L-O mode associated to the terrestrial kilometric radiation."

Page 10, line 28-

"We suggest that the DEMETER kilometric emissions are linked to a Z-mode micro-scale region. This trapping Z-mode region can only be detected between the Earth's ionosphere and the plasmasphere. The hollow cones of this kilometric wave emissions are crossed by the DEMETER orbits at altitudes lower than 700km. Probably the source regions of the DEMETER kilometric emission should be the plasmasphere, like the terrestrial kilometric radiation."

It is still difficult for me to understand your idea of the wave generation and propagation, considering the realistic plasma environment at the plasmapause and in the plasmasphere.

I strongly recommend to add a cartoon as Figure 10, to show your idea of the source mechanism, source location, propagation path with the hollow-cone beam, and observation by DEMETER at low L-shell region.

---

## Referee Report (RR2)

I'm not still convinced with the explanation of the source mechanism, wave mode and propagation pass in Chapter 3. However, in view of the scientific importance of the observational evidence shown in Chapter 2, I decided not to oppose the publication of this article to ANGEO.

Minor comments
1. Figure 3 lower panel;    Horizontal axis is wrong.
2. Figures 7 and 8 are too small.
3. Page 3 line 18;      "between -50 and 0" should be between -30 and 0".
4. Page 3 line 40;    "nigh-side" should be "night-side".
5. Page 4, line 49;    Reference about 'Christmas tree' is needed.
6. Page 7, line 54;    "Fig.8a and 8b" is wrong.
7. Page 8, lines 16 and 19;    Fig9 should be Fig.10.

---

## Author Response (AR2)

**First of all, we thank both reviewers for their constructive comments and suggestions. Our responses are listed below.**

Low altitude banded equatorial kilometric emissions
below the electron cyclotron frequency

Response to Referee #1

*Reviewer1: There are many problems and confusing descriptions in the Sections 3 and 4. (1) Page 7, line 54 "The power level is principally found to increase between 1 and 1.4 L-shell when the magnetic latitude of DEMETER is in between -20◦ and +20◦. The source locations of kilometric wave radiation seem to be confined to a narrow L-shell region."  This means that the source region of DEMETER kilometric wave emission is located in the altitude range from about 6000 to 9000 km above the equatorial upper ionosphere (-20 to +20 MLAT). However, in page 9, line 49 "the beaming of the DEMETER kilometric wave emissions is towards the Earth's ionosphere with sources localized in the plasmasphere." These two sentences are inconsistent.*

A1_(1): The first and the second sentences are may be not correctly written. In the first one, we refer to Fig.7 where the power level is increasing for specific orbital parameters of DEMETER satellite, between 1 and 1.4 L-shell and -20° and +20° in magnetic latitude. In the second sentence, we consider that the beam emitted by the 'plasmaspheric source' is crossing DEMETER trajectories. We have re-considered both sentences in the upgraded version.

*Reviewer1: (2) Page 8, line 48- ."the parallel narrow bands as displayed in Fig.5 are mainly associated to the escaping continuum in the case of the terrestrial kilometric emission." What is the meaning of "associated to the escaping ····". Are they same category waves in the dispersion relation or simply morphological similarity of their spectra?*

A1_(2): We have insisted on the morphological aspects as addressed by the referee. The corresponding part of the text in sub-section 3.2 has been re-written.

*Reviewer1: (3) Page 8, line 58 "AKR-X/INTERBALL-1experiment provided similar emissions particularly in the southern hemisphere at low magnetic latitude and at L-Shell of about 1.2 (Kuril'chik et al., 2001)."  "L-Shell of about 1.2" is wrong, and should be "R/RE of about 1.2". The INTERBALL did not provide the data at low L-Shell region (low altitude and low latitude), but at low R/RE region (low altitude and high latitude).*

A1_(3): We have cited the incorrect reference. L-shell is reported in Kuril'chik, V.N, Boudjada, M.Y, and H.O. Rucker, INTERBALL-1 observations of the plasmaspheric emissions related to terrestrial 'continuum' radio emissions, in Planetary Radio Emissions V, Eds. Rucker, H.O., Kaiser, M.L., and Leblanc, Y., Vienna: Austrian Academy of Sciences Press, 325–335, 2001b. Kuril'chik et al. (2001b) provided the corresponding L-shell of the selected INTERBALL-1 events in Fig.1 and Fig.4. This reference has been added to the Reference Section.

*Reviewer1: (4) Again page 9, line 49 "the beaming of the DEMETER kilometric wave emissions is towards the Earth's ionosphere with sources localized in the plasmasphere." If so, how the waves can propagate to the low altitude equatorial ionosphere through the dense plasma region? What is the propagation mode?*

A1_(4): We have reconsidered this sentence (see A1_(1)). In sub-Section 3.3, we refer to the work of Carpenter et al. (2003) who reported similar source conditions. We have added, in the upgraded version, two other references: Goertz and Strangeway (1995) and Sonwalkar et al (2004).  The propagation of the whistler mode is discussed for the plasmasphere where trapped Z-modes are considered, and later on converted to L-O mode.

*Reviewer1: (5) Page 9, line 51 "It is important to note that the DEMETER kilometric events belongs to a very limited regions in range 1-2 L-Shell (as shown in Fig.7) with distances between 1.1040 RE and 1.1070 RE. This means that the DEMETER orbits is crossing the plasmaspheric hollow cones on few dozen of kilometer."  What is the "hollow cone"? There is no explanation. Physical explanation is necessary. It should be strongly related to the wage generation process.*

A1_(5): We have answered in A1_(1) and in A1_(4) and re-written this part of the text. The 'hollow cone' is the emission diagram associated to the propagation modes (Z- and LO-modes). The cone opening angle is depending on the emitted frequency. Table1 lists the dependence of the opening angle on the emission frequency and the magnetic latitude. Opening angles are found to be of about 5° and 35° at frequencies 400 kHz and 50 kHz, respectively.

*Reviewer1: (6) Figure 9 I suppose that this figure is based on the orbital plasma data from equatorial to polar region where fp, fg, and fz changes greatly. If so, this figure is not applicable for your discussion, because the wave propagation mode to be considered here is restricted to that of the low latitude inner plasmasphere.*

A1_(6): Plasma parameters as discussed in sub-Section 3.3 are those derived from the selected kilometric events. We have given the corresponding equations to derive fp, fg and fz. This means that we did not take orbital plasma data from the equatorial plane to the polar region. In this sub-Section, we have followed the work of Carpenter et al. (2003) who provided a model plot for the Z-mode trapping region.

*Reviewer1: (7) Page 9, line 56 "Probably such restricted regions may be associated to the Z-mode waves which are linked to the free escaping L-O mode as suggested by Jones (1976) in his model. In such region the Z-mode waves are considered to be trapped and later converted into L-O mode associated to the terrestrial kilometric radiation." Revierwer1: Page 10, line 28- "We suggest that the DEMETER kilometric emissions are linked to a Z-mode micro-scale region. This trapping Z-mode region can only be detected between the Earth's ionosphere and the plasmasphere. The hollow cones of this kilometric wave emissions are crossed by the DEMETER orbits at altitudes lower than 700km. Probably the source regions of the DEMETER kilometric emission should be the plasmasphere, like the terrestrial kilometric radiation." It is still difficult for me to understand your idea of the wave generation and propagation, considering the realistic plasma environment at the plasmapause and in the plasmasphere. I strongly recommend to add a cartoon as Figure 10, to show your idea of the source mechanism, source location, propagation path with the hollow-cone beam, and observation by DEMETER at low L-shell region.*

A1_(7): We have answered to referee critics in the above addressed points. In the upgraded version we have added, as suggested by the referee, a schematic representation (i.e. Fig.7 and Fig.8) of the ways the emission beams coming from the plasmasphere interact with DEMETER orbits.

**Response to the Referees**

First of all, we thank both reviewers for their constructive comments and suggestions. Our responses are listed below.

**Low altitude banded equatorial kilometric emissions below the electron cyclotron frequency**

**Response to Referee #2**

*Reviewer2: (1) A paper characterizing these low latitude/altitude emissions below FCE is of scientific interest, however no attempt is made to understand the spacing of the harmonics which highly reduces the potential impact this paper would have. For one of their examples the frequency spacing of the bands is ~25 kHz which is greater than the low hybrid frequency ~20 kHz. Beyond noting the band spacing, no attempt is made to characterize it in this paper. For example, a scatter plot of spacing versus model FLHR, FCE, wave power versus harmonic number etc.*

A2_(1): We totally agree with the referee concerning the frequency intervals as observed for one given event (e.g. Fig.1b and Fig.2b ). However, we have attempted in this paper to emphasize on the statistical approach (occurrence in latitude, intensity level) and the emission beaming of the equatorial kilometric radiation. We did not find in the important previous mission (like IMAGE, GEOTAIL, INTERBALL, CLUSTER) similar spectral features. The reason for this, may be, is the polar low orbits of DEMETER ( ~ 700 km) and the possibility to record the electric field from few kilohertz up to 3.5 MHz.

*Reviewer2: (2) Also the emissions of every ~5th harmonic appears to be broader in magnetic latitude compared to the harmonics between them. Why is this? Could this be a saturation effect of the electric field measurements? Below is an excerpt from the response to the referee reports.*

A2_(2): We did not investigate the frequency intervals and also the harmonic 'components'. We believe that the extension in magnetic latitude for the '5$^{th}$' harmonic is not a saturation effect particularly above 200 kHz. In Fig.1a and Fig.1b the power level above 200 kHz is similar for the '5$^{th'}$' harmonic and also for other harmonics.

*Reviewer2: (3)"A1_A: As suggested by the referee, we have checked and found that the considered kilometric wave emissions can't a free space radiation. In all events the gyrofrequency (FCE) is above the frequency associated to the LF kilometric emission. In Fig.1 or Fig.2 the gyro-frequency is indicated." Since these waves are not free space mode radiation calling them "Kilometric wave radiation" a term used for free space mode radiation generated near the plasmapause is confusing. How about calling them something like 'Low Altitude Banded Equatorial Emissions below the electron cyclotron frequency'.*

A2_(4): We agree with the referee concerning the change of the title. We would like to add the word 'kilometric' to specify the frequency range of such emission, i.e. 50 kHz and up to 800 kHz. The new title is 'Low Altitude Banded Equatorial Kilometric Emissions below the electron cyclotron frequency'

*Reviewer2: (4) I don't see evidence of a continuum in Figures 1&2. Do you mean continuum in time or frequency?*

A2_(4): We use the word 'continuum' to describe the morphological aspects of the narrow bands. This means that the 'continuum' is in time. For a given narrow band, we did not find dis-continuity in the emission, as as one can see in Fig.1 and Fig.2.

*Reviewer2: (5) The introduction and most of the paper focuses on 'free space radiation' not banded emissions below the FCE. The entire paper requires a complete rewrite. As noted in the other referee report, Whistler waves propagate mainly along the field line. The introduction should mainly focus on previous observations of banded emissions below Fce and propagation characteristics of Whistler waves not escaping continuum.*

A2_(5): We have re-written the Introduction Section, as suggested by the referee. However we did not find references concerning similar low altitude equatorial kilometric emissions where the plasma frequency is below the gyro-frequency. Nevertheless we mainly refer in the Introduction to banded emissions observed by Cluster satellites. Also IMAGE investigations are briefly considered where we have insisted on whistler and Z-modes on their relations to specific regions of the plasmasphere.

*Reviewer2: (6) In the discussion section the author should note that that the frequency range and radiation pattern of these observations below FCE is similar to the 'Christmas tree pattern' observed above FCE outside the plasmasphere. Then the authors can speculate on whether this is just a coincidence or if a deeper connection exists.*

A2_(6): We refer in sub-Section 3.2 to the similarity and the discrepancy between the well-known terrestrial kilometric emissions and the DEMETER equatorial emissions. Main references about the terrestrial kilometric emissions are only given in this sub-Section.

*Reviewer2: (7) The Fce line should be labeled in Figures 1 &2.*

A2_(7): The gyrofrequency emission lines, as suggested by the referee, are indicated for both selected events.

**Low altitude banded equatorial kilometric emissions below the electron cyclotron frequency**

[revised manuscript text omitted]

---

## Author Response (AR3)

**Response to the Referees**

**First of all, we are thankful to reviewers for their constructive comments and suggestions.**

**Response to Referee #1**

*Reviewer1: (1) I'm not still convinced with the explanation of the source mechanism, wave mode and propagation pass in Chapter 3. However, in view of the scientific importance of the observational evidence shown in Chapter 2, I decided not to oppose the publication of this article to ANGEO. Minor comments. Figure 3 lower panel; Horizontal axis is wrong.*

    A1_(1): The horizontal axis in Fig.3 (lower panel) has been corrected.

*Reviewer1: (2) Figures 7 and 8 are too small.*

    A1_(2): Sketches of the beaming diagrams for southern (i.e. Fig.7) and northern (i.e. Fig.8) hemispheres have been re-considered, as referee suggested.

*Reviewer1: (3) Page 3 line 18; "between -50 and 0" should be between -30 and 0", and Page 3 line 40; "nigh-side" should be "night-side".*

    A1_(3): Magnetic latitude range has been corrected, and also the miss-typewriting of 'night-side'.

*Reviewer1: (4) Page 4, line 49; Reference about 'Christmas tree' is needed.*

    A1_(4): The Christmas tree has been 'adapted' to emphasize on the spectral patterns of the frequency banded emissions. We decide to write 'tree spectral pattern' instead of 'Christmas tree'. However, in sub-Section 3.2, we cite the 'Christmas tree' and the corresponding reference, i.e. Green and Boardsen (2006).

*Reviewer1: (5) Page 7, line 54; "Fig.8a and 8b" is wrong, and Page 8, lines 16 and 19; Fig9 should be Fig.10.*

    A1_(5): The corrected figure numbers have been corrected in the upgraded version, i.e. Fig.7 (instead of Fig.8a), Fig.8 (instead of Fig.8b) and Fig.10 (instead of Fig.9).

**Response to Referee #2**

*Reviewer2: (1) By definition 'kilometric' refers to distance not frequency and suggests wavelengths on the order of kilometers to 10's of kilometers. Unless you can estimate the wavelength, please drop 'kilometric' from your title and abstract. In the description of the DEMETER events in this paper all references to 'kilometric' should be removed?*

    A2_(1): We agree to cut out the word 'kilometric' from the title, and also from the description Section (i.e. Section 2), as suggested by the referee. This leads to avoid any confusion between the frequency banded emissions and the terrestrial kilometric radiations.

*Reviewer2: (2)These waves might be due to the electrostatic whistler instability see https://arxiv.org/pdf/1707.05346.pdf?*

    A2_(2): A new paragraph has been added in sub-Section 3.3 taking into consideration the paper of An et al. on 'Electrostatic and whistler instabilities excited by an electron beam' (Physics of Plasmas, 24, 072116, 2017).

*Reviewer2: (3) Does DEMETER have an electron instrument?*

    A2_(3): Lebreton et al. (2006) described the 'Instrument Sonde de Langmuir' (ISL) onboard DEMETER satellite. The experiment objectives concern measurements by ISL of the electron

density and the temperature. In our study and for all events, the plasma frequency f_p is, on average, about 100 kHz and can reach 900 kHz, as one can derive from Fig.10. This corresponds to an electron density Ne (cm^-3) between ~10^2 (cm^-3) and 10^4 (cm^-3), using the classical relationship: Ne ~ (f_p/9)^2. Study of Chen et al. (2018), as shown in their Fig.1a, are not 'accurately' comparable to those derived from our investigation. This may be due to the fact that the source regions of the frequency banded emissions are not localized along the DEMETER trajectories.

*Reviewer2: (4) A nonlinear beating (either instrumental or natural) between the whistler mode and LHR near the equator might explain some of the harmonic banding? You should either shelve this paper or include a case study of 1 or 2 events studying the spacing of the bands in frequency and possible explanation. At a time step using the 2 bands with dominant power can you reproduce the other emissions from sums and differences of the frequency of 2 dominant bands?*

A2_(4): Bandwidths of the banded frequencies have been added in sub-Section 2.2. We have estimated such frequency intervals in the case of the two events of Fig.1 and Fig.2. It is not possible to conclude if this effect is artificial and/or natural. Further detailed analysis should be considered.

*Reviewer2: (5) How do you know the emissions are beamed as opposed an electrostatic emission with limited spatial propagation?*

A2_(5): In this paper, we emphasize on the statistical approach. The spectral pattern in Fig.6 is a superposition/overlapping of variation of the emission frequency versus the magnetic latitude. Sketches in Fig.7 and Fig.8 are aimed to provide a representation of the beam dependence (i.e. opening angle) with regard to the magnetic equator. In reality this beam of Fig.6 is the composition of several multi-beams, each one may be associated to one single narrow band. The individual single narrow band should have a limited spatial propagation.

*Reviewer2: (6) Title Low altitude banded equatorial kilometric emissions below the electron cyclotron frequency -> Low altitude frequency banded equatorial emissions observed below the electron cyclotron frequency. Abstract The ICE experiment onboard the DEMETER satellite recorded kilometric wave emissions in the vicinity of the magnetic equatorial plane. Reviewer2: (7) -> The ICE experiment onboard the DEMETER satellite recorded frequency banded wave emissions below fce, with band spacing ~> flhr, in the vicinity of the magnetic equatorial plane. 5. Continuum -> frequency bands continuous in time*

A2_(6): Corrections and suggestions in the Title and the Abstract have been taken into consideration.

**Low altitude frequency banded equatorial kilometric emissions observed below the electron cyclotron frequency**

[revised manuscript text omitted]

---

## Author Response (AR4)

**Response to the Referee**

**We are thankful to reviewers for their constructive comments and suggestions.**

*Reviewer:* **I find the paper acceptable if you indicate in your abstract that then DEMETER emissions are "non-free space", while the "terrestrial kilometric radiation" is free space. Maybe
We show in this study the similarities and the discrepancies between the non-free space DEMETER frequency banded emissions and the well-known free space terrestrial kilometric radiation.**

A: In the Abstract of the paper, we have corrected the corresponding sentence as suggested by the referee.

[revised manuscript text omitted]